# Toxocariasis in a Child with Autism Spectrum Disorder

**DOI:** 10.3390/ijerph18010283

**Published:** 2021-01-02

**Authors:** Filippo Pieroni, Francesco Massei, Maria Vittoria Micheletti, Laura Luti, Emanuela De Marco, Alessandra Ludovisi, Gabriella Casazza, Fabrizio Bruschi

**Affiliations:** 1Postgraduate School of Paediatrics, Department of Clinical and Experimental Medicine, Section of Paediatric, University of Pisa, 56126 Pisa, Italy; 2Pediatric Hematology Oncology, Bone Marrow Transplant, Azienda Ospedaliero Universitaria Pisana, 56126 Pisa, Italy; f.massei@med.unipi.it (F.M.); vittoria.micheletti@gmail.com (M.V.M.); l.luti@ao-pisa.toscana.it (L.L.); e.demarco@ao-pisa.toscana.it (E.D.M.); g.casazza@ao-pisa.toscana.it (G.C.); 3Department of Infectious Diseases, EURLP, Istituto Superiore di Sanità, 00161 Rome, Italy; alessandra.ludovisi@iss.it; 4Department of Translational Research, N.T.M.S., University of Pisa, 56126 Pisa, Italy; fabrizio.bruschi@med.unipi.it; 5Parasitic Disease Monitoring Program, AOUP, 56126 Pisa, Italy

**Keywords:** toxocariasis, *Toxocara canis*, autism, severe eosinophilia

## Abstract

A boy affected by autism spectrum disorder was admitted for persistent high fever, without shiver, for two weeks. The boy referred to abdominal pain, in the first week of fever, and to mild anorexia in the last days before admittance to our hospital centre. The father reported that the boy suffered by geophagia and coprophagia and he has been going to a didactical farm (where he has been exposed to several kinds of animals) to improve his neuropsychiatric condition. Blood analysis shows severe eosinophilia and high levels of total IgE, and abdominal echocardiography showed hepatic lesions. Enzyme-linked immunosorbent assay (ELISA) and Western blot (WB) confirmed the suspicion of toxocariasis, linked to the habit of the boy to ingest ground or animal faeces in a didactic farm frequented by the boy. Treatment with albendazole and prednisone was administered with a rapid improvement of the symptoms and the laboratory findings and significant reduction of the hepatic lesion.

## 1. Introduction

Toxocariasis is caused by the nematode parasite *Toxocara*, which belongs to *Ascaridida* order. *Toxocara canis* and *Toxocara cati* are gastrointestinal parasites of canids and felids, respectively, with a ubiquitous worldwide distribution [1,2].

The most frequent way of transmission for humans is represented by accidental ingestion of ova present in the ground contaminated by dog or cat puppy stool [3]. This risk is more relevant in children who have the habit for pica, as in the case of autism spectrum disorder. A less frequent way of transmission is represented by ingestion of larvae present in organs (liver, for example) from infected hosts such as chicken, cattle and sheep [1]. Once ingested, ova release infecting larvae in the human intestine from which they pass the intestinal wall, then they enter the blood vessels and lymphatic system, to migrate to different organs (liver, lungs, central nervous system, eyes, etc.), leading to visceral *larva migrans* (VLM) or ocular *larva migrans* (OLM) syndromes, respectively. Covert (or subclinical) toxocariasis [4] and the more recently defined neurotoxocariasis (NT) [5] are other possible presentations.

Diagnosis of VLM is based on clinical and image findings, presence of blood eosinophilia and specific laboratory tests, such as an enzyme-linked immunosorbent assay (ELISA), using *Toxocara* excretory-secretory (ES) antigen to be confirmed with WB analysis [1]. 

Here, we present the case of a 7-year-old boy, affected by pervasive disorder of the autism spectrum, with VLM who had positive antibodies response to *Toxocara* ES antigen. 

## 2. Case Report

A 7-year-old child affected by pervasive disorder of the autism spectrum was admitted to our Oncohemathology Paediatric Department with persistent fever of unknown origin (with peaks up to 39 °C, mainly in the evening), without shiver, for about two weeks. The fever was associated with abdominal pain in the first days and he was treated with antipyretics with benefit. Mild anorexia was found in the last days before admittance, and then the parents asked for medical assistance.

When the boy was admitted, the father reported to us that the child has been going to a didactic farm for some years, where he was exposed to several pet animals (dogs and cats, etc.) to improve his neuropsychiatric condition. The boy also suffered by a geophagia and coprophagia, as reported by the farm teachers. The father also reported that 3 years before the admission FE was affected by a knee monoarthitis, treated with benefit with oral non-steroidal anti-inflammatory drugs (NSAIDs). In this last event, mild eosinophilia was found (900/mcl) but no further diagnostic investigations were conducted.

On admission, the boy was febrile (T: 38 °C), but in good general condition. The physical examination did not reveal any alteration of the skin or of the respiratory and cardiac system. No hepatomegaly, no splenomegaly and any enlargement of lymph nodes were found. 

Blood tests showed leucocytosis (36,270/mcL) with severe eosinophilia (26,220/mcL), slight increase of ESR (39 mm/h) and RCP (1.26 mg/dL), hypergammaglobulinemia and high levels of IgG (2370 mg/dL) IgM (236 mg/dL) and total IgE (4980 U/mL). No blasts were found on the peripheral blood smear, ruling out blood diseases. Liver function test, creatinine, glucose and electrolytes were normal. An upper abdominal echography revealed multiple small (less than 10 mm) hypoechoic nodules all over the liver, with some reactive periportal lymph nodes. To confirm these findings, we performed an abdominal computed tomography (CT), which showed hypodense lesions with softly hyperdense peripheral haloes, located in IV, V and VI hepatic segments and sub-centimetre lesions in the left hepatic lobe (see Figure 1A). Chest X-ray examination did not show any involvement of the lungs, but a thoracic CT revealed multiple parenchymal nodules (up to 6.5 mm), located in the peripheral basal segments of the inferior left and the right posterior pulmonary lobes.

In suspicion of parasitic infection, we performed stool examination and serologic evaluation for *Toxoplasma gondii*, *Leishmania spp*, *Taenia solium*, *Echinococcus spp* and *Toxocara spp*. The commercially available ELISA kit for *Toxocara canis* (Nova Tec, srl, Dietzenbach, Germany) revealed an elevated positivity (specificity 98.63%, sensibility 96.92%), which was confirmed by WB (see Figure 2), allowing us to make a diagnosis of Visceral Larva migrans syndrome. Further tests showed an increase of anti-B isohemagglutinin title (1:256). To exclude central nervous system (CNS) and ocular involvement, a brain MRI and an ophthalmologic examination were performed, both resulting negative. Finally, we excluded a cardiac involvement with an echocardiography. 

Once diagnostic evaluation was concluded, treatment with albendazole (15 mg/kg/die orally twice a day, for 5 days) and prednisone (0.8 mg/kg/die for 5 days orally to prevent allergic reaction to parasitic massive lysis) was started. Treatment was well tolerated by the patient without any side effect, with a rapid improvement of the symptoms (disappearance of the fever and appetite recovery) and of the laboratory findings (eosinophilia and IgE reduction, decrease of RCP and ESR). An abdominal echography was repeated after 4 days of therapy, showing significant reduction of the hepatic lesions.

Follow-up was made by repeating blood tests all over 2 years after treatment, showing progressive decrease in eosinophilia and total IgE concentration back to normal value (see Figure 3), and with echografic controls. Almost normal levels of eosinophils and total IgE were reached after a rapid decrease two weeks after the treatment. ELISA test for *Toxocara* continued to show positive results even after 1 year after the treatment, even if blood level of eosinophils and total IgE levels were back to normal values and IL-5 concentration was in range. After 1 month from treatment, abdominal echography was not able to find most of the hepatic lesions previously described, whereas after 1 year only some calcific hepatic lesions were visible (in the same area where the biggest ones where described). A thoracic and abdominal CT done after one year confirmed these finding and showed the disappearance of the pulmonary lesions (see Figure 1B).

At a final control made after 2 years, no sign of infection was found at the blood tests (normal value of eosinophils, ERS, RCP and total IgE levels) and the abdominal echography showed no modifications from the one performed the previous year. ELISA test for *Toxocara* was still positive. An ophthalmologic examination confirmed the absence of local infection. The father reported good conditions of the boy all over the period after the treatment, with no symptoms or signs linkable to the previous *Toxocara* infection.

## 3. Discussion

*Toxocara canis* is an uncommon nematode parasite not frequently observed in clinical daily practise. High incidence of *Toxocara* spp infections are linked to areas of poor hygiene or poverty, other than people living or working with a high contact with infected animals [4,6]. Children usually come in contact with *Toxocara* ova while playing in sandbox or on the ground in places where infected dogs or cats defecate [4,7]. The incidence of this infection is known to be higher in children with geophagia, which is common to find in neuropsychiatric conditions such as autism spectrum disorder [4,8]. Some studies also show that children with reported exposure to *Toxocara* (seropositive patients) have lower grades at cognitive function test than seronegative ones, demonstrating a link between *Toxocara* infection and cognitive impairment [9,10].

Toxocariasis is also linked to the development of immunopathological illness such as asthma and allergic rhinitis after the infection [11,12]. None of these findings were reported in our patient even after 3 years of follow-up. 

Toxocariasis suspicion is made considering anamnestic evaluation, signs and symptoms (which are variable due to localization of granulomas) and laboratory findings (severe eosinophilia). Diagnosis of *Toxocara* infection is obtained by an ELISA test, which confirm exposure to the parasite, combined with WB, which demonstrate the ongoing infection. 

In our case, the didactic farm activities exposed the child with cognitive impairment to an augmented risk of parasitic infection. The hepatic and pulmonary lesions, linked to the clinical presentation and the laboratory findings, allowed to establish the diagnosis of visceral larva migrans. It was difficult to make an optimal neurologic assessment due to the neuropsychiatric disorder of our patient and it was necessary to rule out the CNS (neurotoxocariasis) and ocular involvement (ocular larva migrans) by performing a brain MRI and an opthalmologic examination. 

## 4. Conclusions

Our case demonstrates that neglected infection should not be excluded in differential diagnosis for severe eosinophilia. All the activities practised by the patient should be considered at the first evaluation, particularly those which include close contact with animals. 

Geophagia and coprophagia are common in children with neurological impairment, but the history of accidental ingestion of ground can also be found in healthy patients. These events are often not communicated by the family or by schoolteachers, so we recommend looking for similar events by focused questions during anamnestic evaluation.

Didactical farm or other kind of pet therapy should not be avoided due to the risk of parasitic infection. On the other hand, we suggest performing a screening test for parasite infections in the animal population of every farm connected to a case like that described in the present case report. This might avoid further infections in other children or adults by treating animals with confirmed infection.

## Figures and Tables

**Figure 1 ijerph-18-00283-f001:**
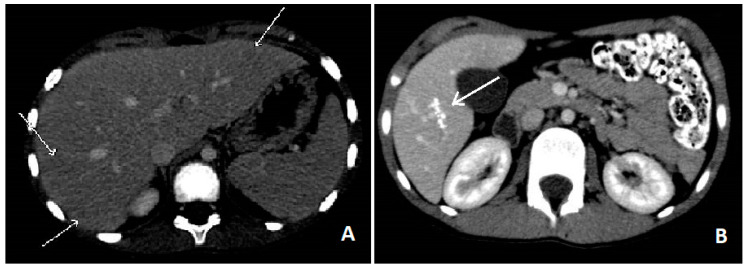
Abdominal CT showing liver hypoechoic nodules due to *Toxocara canis* infection (**A**) and one of the remaining calcific lesions after 1 year (**B**).

**Figure 2 ijerph-18-00283-f002:**
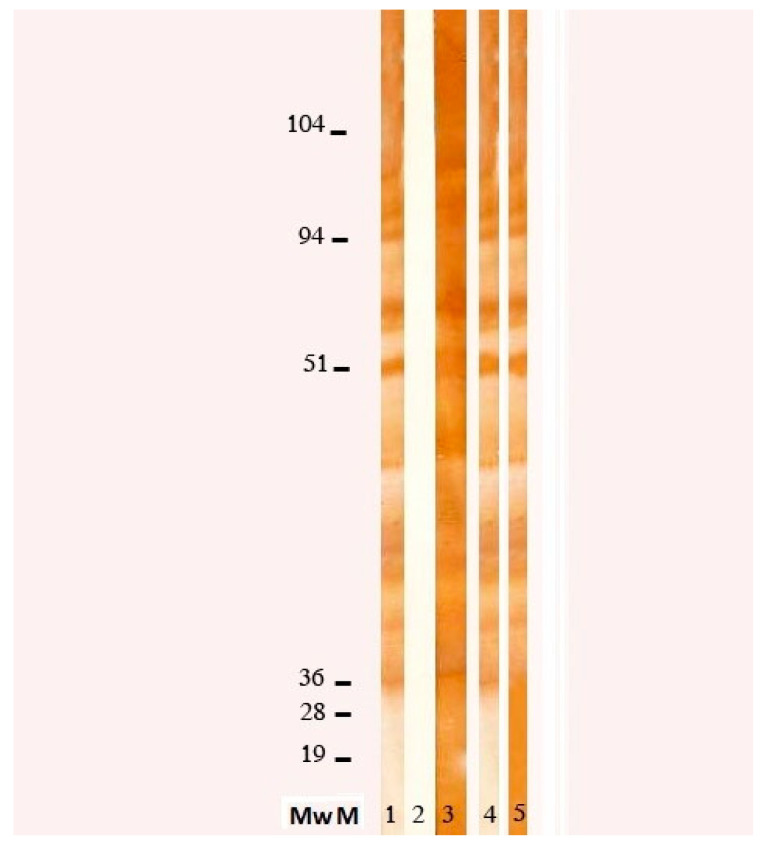
Western blot (WB) pattern of reactivity of *Toxocara canis* excretory/secretory antigens (ESA) with human sera. *T. canis* ESA were subjected to 12% SDS–PAGE in reducing conditions, blotted into nitrocellulose and incubated with human serum samples. MwM = molecular weights in kDa; lane 1, serum sample from a *T. canis* infected human, positive control; lane 2, serum sample from a blood donor, negative control; lines 3, 4 and 5 serum samples from the patient under study collected at 3 different times post-infection: lane 3, sample collected just before the beginning of treatment; lanes 4 and 5, respectively, 4 and 9 months after the end of treatment.

**Figure 3 ijerph-18-00283-f003:**
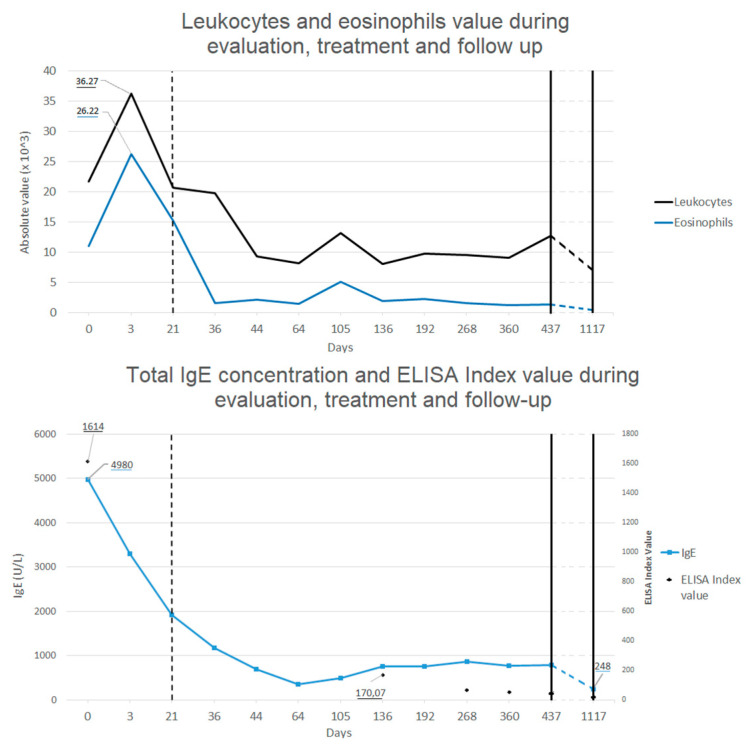
Leukocytes, eosinophils (**upper graphic**), total IgE concentration and enzyme-linked immunosorbent assay (ELISA) index value for Toxocara (**lower graphic**) during evaluation, treatment and follow up. Dotted line marks the beginning of the treatment. Almost 2 years passed between last two blood test.

## Data Availability

The data presented in this study are available on request from the corresponding author. The data are not publicly available due to privacy restriction.

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
