# Peer review of "Toxocariasis in a Child with Autism Spectrum Disorder"

_ijerph, 2021, doi:10.3390/ijerph18010283_

Round 1

Reviewer 1 Report

Authors described a case report of a 7-year-old boy affected by autism spectrum disorder that was diagnosed for Visceral Larva Migrans syndrome. I consider the information presented in this case report an important contribution to scientific community.

The manuscript is well structured and appropriated to be published.

Suggestions and comments to authors:

I suggest that the authors modify the title and remove “and revision of the literature”  I would expect more information to consider this manuscript as a revision of the literature

I suggest that the authors use the word “egg” instead “ova”

Line 34: Toxocara canis can infect other canids and T. cati can infect other felids. I consider the words “canids and felids” more appropriate to replace “dogs and cats”

Lines 85-85: I suggest authors to review this sentence “… we performed stool examination and the detection of specific antibodies for…” In fact the serological test ELISA can detect specific antibodies for this parasites, but the way it is written seems that authors detected antibodies against these parasites.  The antibodies were detected only in positive test (ELISA for Toxocara spp).

Line 85: Taenia solium instead Taenia solum. I suggest to insert gondii after Toxoplasma and spp after Leishmania, Echinococcus and Toxocara.

It is important to mention the sensitivity and specificity of ELISA used to detect antibody against Toxocara spp.

Line 87: authors mention that the specificity was confirmed by WB. I did not understand the meaning of this sentence since specificity is related to the “true negatives”.

Figure 2: Caption of figure 2 quote “Mw, M” in line 94, but it does not appear in the figure.

Line 134: Italicize “Toxocara”

Author Response

First, we would like to thank the reviewer for consideration and help with our article. We gladly accepted most of the suggestion made and we would like to respond point by point to the consideration made:

  1. Regarding the title, we understood that title could be misleading. We decided to change it, considering the suggestion from another reviewer, into “Toxocariasis in a boy with autism spectrum disorder: a case report and review of the literature”. We hope that this title makes it clear that our aim is not to change what can be found in literature about Toxocariasis.
  2. We agreed with you to change most of the terms you pointed us (like “ova” instead of “eggs”). We also agreed to change line 34 (“canids and felids”), because your suggestion made it less specific and more appropriate than “dogs and cats”, and line 85, to avoid misleading of the sentences.
  3. We found the characteristic (specificity and sensibility) of the ELISA kit for Toxocara spp used at our laboratory center and we wrote them on line 87
  4. Figure 2 was also corrected to permit a simpler comprehension.

We hope that all the corrections made will please you and the other reviewer. We will wait for further suggestions.

Sincerely,

Dr Filippo Pieroni

Reviewer 2 Report

Pieroni et al. present a case report of toxocariasis in a 7 y.o. male. The proper diagnosis by ELISA and western blot following the report of previous conditions that suggested the infection by some kind of parasite, represents an important protocol that should be followed by other physicians facing similar cases. Nevertheless, some concerns need to be addressed before I can recommend the publication of this report.

1-The text would benefit from a proof-reading by a native speaker to correct some mistakes such as:

  • "The boy referred to abdominal pain.."
  • "ELISA and western blot"
  • Some acronyms are not explained, e.g. FE and CNS

2-The report over-emphasizes the condition of the patient in the autism spectrum disorder (ASD); however, the direct cause of the infection is the geophagia and coprophagia of the patient, which can be attributed to the ASD condition. In other words, the ASD is an indirect cause that create part of the circumstances for the infection.

3-In figure 3, there is a clear trend of reduction by the time that the treatments started. It is worth explaining if any other intervention were performed in the 21 days preceding the start of the treatments, otherwise the graph is suggesting without the treatment the infection could have continued its trend and disappeared by itself.

4-Lines 149-152, it is unclear why the "neuropsychiatric disorder of the patient" is a reason to require testing the CNS and ocular involvement. In other words, it is expected that independently of the ASD condition, any patient with toxocariasis should be tested to rule out the CNS and ocular involvement, then why is the ASD condition singled out to justify these tests?

5-There is a big opportunity area in the discussion or conclusion to contextualize this case for future cases. Such as the risk for patients with geophagia visiting didactical farms or the need for further actions such as notifying the farm to treat the animals and avoid further infections.

6-An ethical note should be included to inform readers whether this study counts with the consent of the family to publish the report, oversight by any ethical committee or the reasons why it doesn't need such ethical approval.

Author Response

First, we would like to thank the reviewer for consideration and help with our article. We gladly accepted most of the suggestion made and we would like to respond point by point to the considerations made:

  1. We did further analysis of the text and we corrected some language mistakes done during the editing process.
  2. We would like to point out that we considered the neurologic impairment of our patient in all its aspects (especially geophagia), and not just by the ASD. However, we decided to focus on this aspect in the new “conclusion” chapter that we had to add to the case report. In this chapter, we decided also to focus on your other suggestions for our work.
  3. 21 days passed from the first blood test and the beginning of toxocariasis treatment in our patient. During this period we performed a second hematic analysis to confirm the severe eosinophilia and to do further analysis (like the ELISA test for Toxocara spp). We performed a third haematic analysis the day treatment began (day 21st). Due to this fact, and that no other kind of treatment was administered to the boy during this period, we considered the values found like normal fluctuation of eosinophil cells count during toxocariasis. If your opinion is that we should be more specific about those values, we could add a small paragraph under the figure to explain that. Also, we could point out that total IgE count is not linked to toxocariasis treatment.”)
  4. Talking about line 149-152 and as explained at point 2, it was difficult to do a neurological assessment due to the neuropsychiatric disorder. We were forced to do a brain MRI and an ophthalmologic evaluation to exclude neurotoxocariasis and ocular larva migrans. However, we accepted your suggestion and decided to point out this difficulty in the new version of the report.
  5. Finally, we added a paragraph to confirm that we had written consent from the parents to publish this report.

We hope that all the corrections made will please you and the other reviewer. We will wait for further suggestions.

Sincerely,

Dr Filippo Pieroni

Reviewer 3 Report

It is a well described case report for a parasitic infection in a boy. Since many other parasitic infections are considered mainly linked to poor countries and poor hygienic conditions and almost ignored in developed countries,it is important do notify this case. Your case demonstrates that some parasitic infection should be evaluated in presence of behavior and nutrition abnormalities. Pet and farm animals therapy is increasingly diffused in developed countries due its potential  efficacy in some neurological diseases in elders and of course in children with abnormal behavior. 

It could be possible that other similar cases can happen and might be considered when pet therapy is performed to people with cognitive impairment and especially in natural environment. So, maybe you should make some more comments and some references related to didactic farms and/or animal therapy

also, suggesting that parents and/or relatives should mention this activity to the doctors for anamnesis when unusual infectious diseases are suspected. English should be a little revised, although I am not an English native speaker.

Author Response

First, we would like to thank the reviewer for consideration and help with our article. We gladly accepted your suggestions. We provided to add a “ conclusion” chapter where we focus on the rule of a good anamnesis in the diagnostic process and the rule of a place like didactical farm in this kind of parasite infection.

We hope that all the corrections made will please you and the other reviewers. We will wait for further suggestions.

Sincerely,

Dr Filippo Pieroni